# Pharmacokinetic–Pharmacodynamic Modeling of Midazolam in Pediatric Surgery

**DOI:** 10.3390/pharmaceutics15112565

**Published:** 2023-11-01

**Authors:** Carmen Flores-Pérez, Luis Alfonso Moreno-Rocha, Juan Luis Chávez-Pacheco, Norma Angélica Noguez-Méndez, Janett Flores-Pérez, Delfina Ortiz-Marmolejo, Lina Andrea Sarmiento-Argüello

**Affiliations:** 1Pharmacology Laboratory, National Institute of Pediatrics, Mexico City 04530, Mexico; afloresp@pediatria.gob.mx (C.F.-P.); jchavezp@pediatria.gob.mx (J.L.C.-P.); mfloresp@pediatria.gob.mx (J.F.-P.); 2Pharmacokinetics and Pharmacodynamics Laboratory, Division of Biological and Health Sciences, Universidad Autónoma Metropolitana, Mexico City 04960, Mexico; 3Molecular and Controlled Release Pharmacy Laboratory, Division of Biological and Health Sciences, Universidad Autónoma Metropolitana, Mexico City 04960, Mexico; nanoguez@correo.xoc.uam; 4Anesthesiology Department, INP, Mexico City 04530, Mexico; dortizm@pediatria.gob.mx (D.O.-M.); lsarmientoa@pediatria.gob.mx (L.A.S.-A.)

**Keywords:** midazolam, PK-PD population model, sedation, pediatrics, minor surgeries

## Abstract

Midazolam (MDZ) is used for sedation in surgical procedures; its clinical effect is related to its receptor affinity and the dose administered. Therefore, a pharmacokinetic–pharmacodynamic (PK-PD) population model of MDZ in pediatric patients undergoing minor surgery is proposed. A descriptive, observational, prospective, and longitudinal, study that included patients of both sexes, aged 2–17 years, ASA I/II, who received MDZ in IV doses (0.05 mg/kg) before surgery. Three blood samples were randomly taken between 5–120 min; both sedation by the Bispectral Index Scale (BIS) and its adverse effects were recorded. The PK-PD relationship was determined using a nonlinear mixed-effects, bicompartmental first-order elimination model using Monolix Suite™. Concentrations and the BIS were fitted to the sigmoid Emax PK-PD population and sigmoid Emax PK/PD indirect binding models, obtaining drug concentrations at the effect site (biophase). The relationship of concentrations and BIS showed a clockwise hysteresis loop, probably indicating time-dependent protein binding. Of note, at half the dose used in pediatric patients, adequate sedation without adverse effects was demonstrated. Further PK-PD studies are needed to optimize dosing schedules and avoid overdosing or possible adverse effects.

## 1. Introduction

Midazolam (MDZ) is one of the drugs of choice for the induction and maintenance of general anesthesia, as well as for premedication before surgical surgery, because it produces anxiolytic, sedating effects, in addition to being an anesthetic adjuvant [1,2,3]. Among its advantages, it has a short half-life, minimal hemodynamic effects and mild respiratory depression [4]. During surgical interventions, it is necessary to be extremely cautious with the dosage of drugs such as MDZ because in cases of overdosage respiratory compromise and central nervous system depression may occur [5].

MDZ is well absorbed intramuscularly, orally, rectally or intranasally. It is highly lipophilic (Figure 1) and rapidly crosses the blood–brain barrier, binding to benzodiazepine receptors in the CNS; onset of action can be 1.5–15 min [6,7], sedation occurs between 20–40 min [8], has a steady-state volume of distribution ranging from 0.8 to 1.7 L/kg, binds 96–98% mainly to albumin and peak serum concentrations can be reached from 23 to 40 min. It has extensive hepatic metabolism, mediated by CYP3A4 and CYP3A5, and its active metabolite is 1-hydroxymidazolam [9,10,11]. From 60 to 80% of the dose is excreted in the urine as glucuronic acid-conjugated alpha-hydroxymidazolam, and less than 1% is eliminated unchanged [12]. The clearance is 6–11 mL/kg/min, and the elimination half-life ranges from 1.5 to 3 h with an action of 60–120 min [10,13,14].

The toxicity of midazolam is rare, but it may occur when combined with other CNS depressants, such as alcohol, opioids and other tricyclic antidepressants. The risk of toxicity increases with intravenous administration and in elderly people with COPD. Ataxia, nystagmus, hypotension, slurred speech, impaired motor coordination, coma and death are some of the symptoms that occur due to midazolam overdose. Flumazenil and activated charcoal are suitable compounds to counteract midazolam toxicity, the latter being an option within one hour after intoxication [15].

Pharmacokinetic–pharmacodynamic (PK-PD) modeling is widely used for the analytical approach of clinically important variables regarding drug concentrations and their pharmacological effect. The PK-PD model identifies drug properties in vivo, which allows characterization and prediction of the time course of drug effects under physiological and pathological conditions [16]. The use of PK-PD models allows the individualization of the parameters obtained, which can be applied directly to the patient considering his or her altered physiological state, making it possible to stop using titration and incorporate the control of the administered dose and the concentration at the site of effect into the clinic [17]. They are also a tool for the design of dosing regimens, optimization and rational use of drugs [16,18,19].

In two PK-PD trials in healthy volunteers, it was found that there was a delay between the plasma concentration reached and the observed clinical effect, producing a hysteresis loop [20,21]. The mechanisms proposed to explain this phenomenon are delay in the equilibrium between the drug concentration in plasma and at the effect site, as well as a delay in the equilibrium at the effect site and in the time to onset of the peak effect [22].

As a short-acting sedative-hypnotic drug, MDZ has been frequently used in children before surgery to minimize preoperative anxiety and facilitate the induction of anesthesia [23]. Children differ from adults in their response to drugs, while this may be the result of changes in dose exposure (PK) and/or presentation of response (PD), the magnitude of these changes may not be reflected solely by differences in body weight [24], age, nutritional status or chronic diseases [25]. An advantage of these models is the ability to establish the clinical relevance of factors known to affect pharmacokinetics or pharmacodynamics in children [26].

Previous results from our group [27] showed that children younger than 6 years would probably require an intravenous dose of less than 0.05 mg/kg to achieve deep sedation, whereas those older than 6 years may require a second dose of MDZ if surgery is prolonged for 120 min or more; likewise, male patients may require a higher dose to achieve deep sedation.

For the above reasons, the present study aimed to determine the pharmacokinetic–pharmacodynamic behavior of MDZ in a cohort of pediatric patients undergoing minor surgery to achieve adequate sedation, and avoid overdosage and possible adverse reactions.

## 2. Materials and Methods

### 2.1. Classification of the Study

Descriptive, observational, longitudinal, prospective study. Protocol INP 012/2019 was conducted following the Helsinki Declaration and approved by the Committees of the National Institute of Pediatrics (INP): Research, Biosafety and Ethics (IRB 00013674).

### 2.2. Patients

A cohort of patients aged 2 to 17 years, weighing more than 10 kg, of both sexes, undergoing minor surgery, from pediatric surgery, plastic surgery, orthopedics, otorhinolaryngology, ophthalmology and urological surgery, and with American Society of Anesthesiologists physical status classification in the anesthetic risk assessment in patients who will undergo surgical events (ASA), ASA I (healthy patient requiring surgery with no history or added pathology) and ASA II (patient with systemic disease, but compensated) were included [28]. They were scheduled for minor surgeries, and were administered an intravenous (IV) dose of MDZ 0.05 mg/kg (PiSA^®^ Pharmaceuticals, Guadalajara, Jal., Mexico) as a sedative before the surgical procedure, during the period from March 2019 to March 2021. Parents, progenitors or guardians signed informed consent, and patients over seven years of age signed the informed assent form.

The patients were stratified by age into three groups: 2–5 years (preschoolers), 6–12 years (schoolchildren) and 13–17 years (adolescents). Both hospitalized patients, who had peripheral venous access, and outpatients, who were referred to the operating room, were checked for compliance with all documentation, including registration forms, privacy notices, informed consent and/or assent forms. Blood samples were taken randomly, 3 times per patient, noting in each sample the time value corresponding to the degree of depth of sedation using the Bispectral Index (BIS) scale, for which a pediatric BIS™ sensor and a BIS™ monitor model A-2000 (Covidien, Mansfield, MA, USA) were used. Based on an interval guide, degrees of depth can be displayed according to the BIS value; a BIS value close to 100 indicates that the patient is unawake, while a BIS value of 0 indicates the absence of brain electrical activity. BIS values less than 60 indicate deep sedation [29].

The institutional format was used to record the possible adverse effects that could occur during treatment with MDZ, which is under the Mexican Official Standard NOM-220-SSA1-2016 about the installation and operation of pharmacovigilance [30].

#### Sample Size

The calculation was performed with the statistical program G power 3.1.9.7 (Universität Kiel, Kiel, Germany), using the a priori analysis: compute needed sample size, with the following parameters: effect size (f = 0.2); significance level (α = 0.05); power (1 − β = 0.9); groups = 3 and number of measurements = 10, resulting in a total sample size of 60 patients (20 patients per group).

### 2.3. Samples

Blood samples (three per patient) were collected in 3 mL Vacutainer™ tubes with heparin (Becton Dickinson, Franklin Lakes, NJ, USA) then labeled and randomly assigned based on a randomization table of sampling times for each age group using envelopes containing the patient number and corresponding sampling times (5, 10, 15, 20, 25, 30, 45, 60, 90 and 120 min). The samples were transferred from the Department of Anesthesiology to the Pharmacology Laboratory, where they were stored at −80 °C until analysis.

Aliquots of 50 μL of liquid blood were taken from the Vacutainer tubes and transferred to Guthrie card circles. Blood spots on the cards were allowed to dry at room temperature and were protected from light for 6 h. Subsequently, the cards containing the dried blood spots (DBS) were stored at −80 °C until analysis.

The determination of MDZ concentrations in dried blood samples (DBS) and the evaluation of BIS in patients are described in detail in Flores-Pérez et al. (2023) [27].

### 2.4. Data Collection

The information was collected in a format containing the patient’s general data, the pharmacokinetic parameters to be evaluated and the data on the level of sedation using the BIS, as well as an institutional format for reporting adverse drug reactions based on the guidelines established by the Pharmacy and Therapeutics Committee of the INP.

### 2.5. Population PK Analysis

A population PK approach with a nonlinear mixed-effects model was carried out using Monolix software (Monolix Suite™ 2021R2, Antony, France, http://www.lixoft.com (accessed on 23 August 2023)), which combines a stochastic expectation maximization algorithm and Markov chain Monte Carlo procedure to maximize the likelihood [31,32].

### 2.6. Building the Model

#### 2.6.1. Structural Model

Midazolam concentration–time data were described by compartmental PK modeling. We analyzed 1- and 2-compartment models with zero-order insertion and first-order elimination. Model fit was visually verified by generating diagnostic plots. In addition, models were selected based on the accuracy of parameter estimates, Bayesian information criterion (BIC) reduction, between-subject variability (BSV) and prediction-corrected visual predictive check (pc-VPC) plots.

#### 2.6.2. Interindividual and Error Models

Interindividual variability in PK parameters was ascribed to an exponential model according to the equation:(1)$$\theta_{j}=\theta_{p}\times e\left(\eta_{i}\right)$$
where $\theta_{j}$ is the model-predicted PK parameter estimate for the jth patient, $\theta_{p}$ is a typical population PK parameter and $\eta_{i}$ is a random variable of normal distribution with mean zero and variance *ω*_2_, which is an estimate. The covariance *η* between two elements is a measure of the statistical relationship between variables (e.g., Cl–V1, Cl–Q, where Cl is the clearance, V1 is the central volume of the distribution volume, and Q is an intercompartmental flow rate); its correlation was estimated from the elements of the variance–covariance matrix according to the following equation:(2)$$R=covariance/\sqrt{\left({\omega_{\theta 1}}^{2}+{\omega_{\theta 1}}^{2}\right)}.$$

The likelihood ratio test, which included log likelihood, Akaike’s information criterion and BIC, was used to test different hypotheses about the structure of the variance–covariance matrix of the BSV parameters. Residual variability, which includes intraindividual variability, measurement errors and model errors, was estimated using the additive and relative error models.
(3)$$C_{ij}=P_{ij}+\varepsilon \;\mathrm{and}\;C_{ij}=P_{ij}\left(1+\varepsilon_{p}\right)$$
where $C_{ij}$ and $P_{ij}$ are the observed and predicted concentrations of midazolam for the jth patient at time *i*, ε and $\varepsilon_{p}$ represent the error, a random variable with a normal distribution, and zero mean and variance $\sigma^{2}$ and $\sigma_{p}^{2}$.

### 2.7. Population PK-PD Analysis

To build the PK-PD models, we used a sequential approach, which consisted of first building the PK model and then keeping the PK parameters fixed with individual estimates, and the parameters of the PD model were estimated.

The mean BIS as a measure of pharmacodynamic effect was related to MDZ concentrations by means of a maximal immediate response inhibition model; if the resulting curve exhibited a hysteresis loop, a pharmacokinetic model linked to an effect compartment was used to collapse the hysteresis as described by Holford and Sheiner (1981) [33], based on equations:(4)$$E=E0\times \left(1-Imax\cdot Ce/\left(Ce+IC50\right)\right)$$
(5)$$E=E0\times \left(1-Imax\cdot Ce^{\gamma }/\left(Ce^{\gamma }+IC50^{\gamma }\right)\right)$$
where *E*0 is the baseline effect, Imax is the maximal fraction of inhibition, *IC*50 is the half-maximal inhibitory concentration, *Ce* is the effect-compartment concentration and *γ* is the Hill exponent that describes the steepness of the *Ce* versus effect curve.

### 2.8. Model Evaluation

The model’s fit was visually inspected by generating diagnostic plots and pc-VPC plots. Models were further selected based on the decrease in parameter estimates: −2 × log-likelihood (objective function value, OFV), Akaike information criterion (AIC), Bayesian information criterion (BIC) and corrected Bayesian information criterion (BICc).

### 2.9. Statistical Analysis

The statistical analysis of the data was performed with Minitab^®^ version 19.2020.1 (Minitab Inc., State College, PA, USA). The point estimators of the parameters were reported with a confidence interval of 95%. The level of statistical significance was set at *p* ≤ 0.05 for all main effect statistical tests.

## 3. Results

### 3.1. Method for MDZ Quantification in Dried Blood Spots

The analysis of the samples was based on a method previously reported in our laboratory [34] with some modifications, and the parameters for the quantification of drug concentrations in DBS were revalidated. The method proved to be reliable, showing linearity in the range of 10–1000 ng/mL, with correlation and determination coefficients of r = 0.9999 and r^2^ = 0.999, respectively, and inter- and intraday coefficients of variation <10%.

### 3.2. Patient Data

Out of a total of 102 pediatric patients who agreed to participate in the study and met the inclusion criteria, only 97 fit the model, which included 59 males and 38 females stratified into three age groups: preschoolers (n = 26), schoolchildren (n = 40) and adolescents (n = 31). Notably, no adverse effects were reported in any patient. The demographic and clinical characteristics of the patients are shown in Table 1.

### 3.3. Midazolam Concentrations and Degree of Sedation Depth

Figure 2 shows the time course of MDZ concentrations and the degree of depth of sedation using the BIS scale in the different age groups and the total number of pediatric patients undergoing minor surgery.

There is great variability in MDZ concentration profiles due to age (Figure 2), especially in the 2- to 5-year-old group, where high levels are observed from 60 min, compared to older patients, where the elimination phase of the drug is observed from 30 min. Regarding the total group of patients, it can be observed that the mean MDZ concentration presents an elimination phase from 30 min, but is not so pronounced and remains almost constant from 45 min. In Figure 2 (right), the BIS behavior is similar among the groups, being found within the deep sedation interval (BIS values between 40 and 60, dotted green lines) from 5 to 60 min; subsequently, preschoolers remain under deep sedation until 120 min, while in the groups of schoolchildren, adolescents and the total group of patients, it is observed that from 90 min onwards, they pass into a phase of moderate sedation, indicating that they are close to awakening.

### 3.4. Pharmacokinetic Model

One- and two-compartment models were used for the evaluation of MDZ PK. A two-compartment open model with linear elimination described the temporal course of MDZ concentrations better than a one-compartment model (Bayesian Information Criterion [ΔBIC]). The pharmacokinetic variables of clearance (Cl), central compartment volume of distribution (V1), intercompartmental flow rate (Q) and peripheral compartment volume of distribution (V2) were calculated for MDZ concentrations up to 120 min in all patients (Table 2).

The residual variability was best described by a proportional model. The individual values of predicted versus observed MDZ concentrations were judged as acceptable, and the trend line was close to the line of the unit (Figure 3). The normalized prediction distribution error (NPDE) analysis revealed that the points were distributed symmetrically with respect to the zero line and that all points were between −3 and +3 units, as shown in Figure 4A,B.

There was some evidence of bias because NPDE fits partially well in a symmetric distribution (Figure 4C). The pc-VPC plots for MDZ concentrations are shown in Figure 5. In total, 20% of the observed concentrations fell outside 10%, and 30% fell outside the 50% confidence interval of the simulated data.

### 3.5. MDZ Pharmacodynamic Model

Sigmoidal Imax and Imax models were tested, and an Imax model related to the baseline effect (*E*0) was found to best describe the pharmacodynamic profiles. When the BIS was plotted against blood concentrations, the resulting curves exhibited a clockwise hysteresis loop (Figure 6C), which may occur as a consequence of different pharmacokinetic and pharmacodynamic mechanisms.

### 3.6. Pharmacokinetic-Pharmacodynamic Model (PK-PD)

Hysteresis collapse was achieved using Equation (4), and the observed BIS was related to *Ce* using the *Imax* model. The pharmacodynamic parameters obtained with this fitting are listed in Table 3.

Figure 7 shows that the effect data were described as a function of the estimated *Ce* values.

## 4. Discussion

In the absence of pediatric pharmacokinetic studies to guide the safe and effective use of drugs, pediatric dosing can be guided by knowledge of anatomical and physiological factors, which help us to understand drug disposition and the influence of these factors in this population [37].

The anatomical, physiological and biochemical changes that occur from birth, as well as the processes of development and maturation in children, condition the disposition of drugs, so it is necessary to perform specific studies in different pediatric ages to establish dosing schemes in the pediatric population [38].

In the Anesthesiology Department of the INP, a dose of MDZ of 0.1 mg/kg is usually administered, which generates deep sedation with BIS values between 28 and 32 (personal communication), so, in this study, a decrease to 0.05 mg/kg was proposed.

Most of the published data are related to trials in volunteers (early validation studies) and patients in the operating room. The publications show that the BIS performs adequately in the measurement of some of the sedative effects of drugs [39,40,41,42].

In the analysis of the time course of MDZ concentrations and sedation levels, different behaviors were observed among the different age groups. In school and adolescent subjects, the elimination phase of the drug started after 30 min, while in preschoolers, the elimination phase was not evident in the sampling period (Figure 2). The BIS values (initial values between 40–60) showed that the dose administered was adequate to achieve deep sedation in all patient groups (Figure 2); however, there were differences in the final phase of the study. School and adolescent patients had a BIS of 60 or more from 60 min (indicating the process of awakening from sedation), while preschool patients remained in deep sedation until 120 min (BIS < 60). This is probably because younger children are more susceptible to the sedative effect due to their immaturity in the processes of absorption, distribution, metabolism and excretion [43] and, therefore, take longer to wake up compared to older children.

It is also known that physiological differences between children and adults can lead to age-related changes in pharmacokinetics and pharmacodynamics. Differences in drug absorption and first-pass metabolism have been described in children compared to adults. These differences may be due to a smaller intestine in children and altered permeability; factors such as gastric pH and emptying time, intestinal transit time, immaturity of secretion, bile and pancreatic fluid activity and blood flow to the intestines and liver may be altered in children, and the activity of intestinal and hepatic drug-metabolizing enzymes may be different from that of adults [38,44]. Other contributing factors could be membrane permeability, binding to plasma proteins and total body water. In drug metabolism, significant differences have been found in the pediatric population compared to adults for both phase I and phase II metabolic enzymes. The immaturity of glomerular filtration, renal tubular secretion and tubular reabsorption at birth and their maturation determine the different drug excretion in the pediatric population compared to adults [38]. Midazolam may be more effective in children because it has a faster onset of action and shorter duration than other benzodiazepines [45].

In the observed relationship of effect as a function of midazolam concentration, a clockwise hysteresis loop was shown (Figure 6B), i.e., time-dependent protein binding. Hysteresis loops suggest that the relationship between drug concentration and the effect being measured is not a simple direct relationship but may have an inherent delay and imbalance, which may be the result of metabolites, the consequence of changes in pharmacodynamics or the use of a nonspecific assay, or they may imply an indirect relationship.

In a simulation study, time-dependent protein binding can occur as a consequence of a time-dependent decrease in protein concentration in serum by the displacement of a metabolite. When the pharmacological effect was plotted versus the total drug concentration in serum, a clockwise hysteresis was evident; however, when the concentration of free drug in serum, which was correlated with the pharmacological effect, was plotted, no hysteresis was evident [19]. Time-dependent protein binding that can occur because of an increase in protein concentration in serum can lead to a decrease in the free fraction of the drug. Despite these theoretical simulations and modeling, no examples of studies in the literature explaining this phenomenon are apparent to date.

PK-PD models are a clinical pharmacology tool to optimize drug use by designing rational dosage forms and dosing regimens. The quantitative representation of the dose concentration–response relationship should provide information for the prediction of the level of response at a given drug dose level, taking into account underlying physiological processes such as disease state, age, sex [16] and factors such as weight, ethnicity, genotype and others that may influence drug clearance [46].

Cl and Vd parameters in our patients had low values, which could be explained by some factors: (a) Differences in CYP3A4 expression and activity in the liver and intestine produce a lower clearance of midazolam in children than adults [47,48]. (b) The presence of single nucleotide polymorphisms in midazolam metabolism genes, e.g., the CYP3A4*1B variant, is associated with low enzyme expression and activity, while CYP3A5*3 is associated with lower enzyme activity and lower drug clearance, with potential effects on drug levels and/or toxicity [49,50,51,52,53,54]; and (c) concomitant drugs that are metabolized by the CYP3A pathway, such as propofol, fentanyl, lidocaine, dexamethasone and paracetamol. A relevant finding in this work was to demonstrate that Imax values were within the range of deep sedation (BIS 40–60) even when half the usual dose was administered in the pediatric population undergoing minor surgeries at the INP. This is important, as it guarantees the sedative effect and decreases the presence of adverse reactions and overdosage of MDZ.

Sun et al. (2008), in a study in adults, recommend that a decrease in the dose of midazolam may be clinically effective [4]; this is consistent with the results obtained in the group of preschoolers (2–5 years), where we observed a prolonged sedative effect up to 120 min after the administered dose, resulting in patients taking longer to awaken from anesthesia. In this sense, the pharmacokinetics necessary to understand the action of a drug may not be closely related to the pharmacodynamic response. Although drug concentrations decrease rapidly, binding of MDZ to the brain and other receptors may result in an action disproportionate to the half-life of the drug [55]. Our results suggest that children < 6 years may require lower doses of MDZ (<0.05 mg/kg) for adequate sedative effects, whereas children > 6 years may require a second dose to maintain sedation in surgeries longer than 120 min.

In the pediatric population, some drugs have a pharmacodynamic behavior different from that of adults, which determines particularities in their therapeutic effects, as is the case for sedatives [56]. The commonly used method to evaluate the depth of anesthesia is the electroencephalogram, or a modification of the signals detected by it (spectral edge frequency, Biespectral, etc.) [57]. The BIS showed a close relationship with the modeled propofol concentration at the effect site and serves as a measure of the anesthetic drug effect in children older than 1 year [58]. Several clinical studies on the efficacy of oral midazolam in children did not identify significant age effects [59,60]; for example, a multicenter study (397 children) found that the efficacy of midazolam was not age-dependent and that a dose of 0.25 mg/kg was as effective as higher doses. These data agree with our results, in which we did not appreciate differences in the depth of sedation by BIS in the age groups studied. The IV. dose of 0.05 mg/kg midazolam administered to the patients was sufficient to obtain adequate sedation (BIS values between 40–60) in all three age groups, with no change in the drug clearance rate.

In children, Vd changes with age, and such changes are due to body composition (especially extracellular and total body water volumes) and plasma protein binding [61,62]. Gender differences in drug disposition also become more evident during puberty [63,64,65]. The effects of rapid growth, sexual maturation and a large variability in pharmacokinetics should be considered during studies with adolescents [61,64,66]. Determination of the bolus dose to achieve a target concentration is dependent on Vd and increases with decreasing age [67]. This is in agreement with the results of our study, since we observed that Vd1 values increased as age decreased in both sexes. According to Vaughns et al. (2018), this could be due to children from one year of age onward having a higher percentage of fat and a lower percentage of protein (22.4% fat and 13.4% protein) compared to older children (13% fat and 18.1% protein) [62].

Sex differences in drug metabolism may have an important influence on pharmacokinetics [68]. Some studies indicate that women metabolize drugs more rapidly than men. This is particularly the case for substrates of the major metabolic cytochrome P450 enzyme, CYP3A4, which may explain one of the mechanisms by which there may be a difference between Vd and drug concentration between male and female subjects [37].

The differences found in Vd and BIS in preschool patients concerning schoolchildren and adolescents may be due to immature CYP3A activity and body fat content. This is due to changes in body composition and protein binding. A decrease in binding, due to a lower protein percentage, may increase the Vd of midazolam, which is already high in these patients, by allowing more rapid and extensive redistribution. Changes in the size of biological fluid compartments and alterations in regional blood flow and membrane permeability may also alter the speed and extent of redistribution. Body fat, which will bind to a fat-soluble drug, increases with age and will increase the volume of distribution [55]. The latter is consistent with the peripheral V2 compartment Vd values we obtained in male patients, which increased with age. Although there is evidence that fat-soluble drugs, such as midazolam, are more widely distributed in women than in men [69] because women are smaller and have more fat, less muscle and less body water [70], in our study, we could not appreciate this, perhaps because we did not have the same number of male and female subjects (male sex predominated in all age groups) to observe the differences between sexes.

With respect to the ethnic origin of the patients, all were of Mexican nationality, with characteristic “mestizo” features; however, we cannot accurately determine the ethnic origin since we did not perform origin surveys of grandparents or great-grandparents or analysis of molecular markers of ancestry.

With regard to side effects, it should be noted that we did not obtain any reports on the duration of the surgical procedure. A limitation of the study was that we did not consider following up the patients in the postsurgical stage, especially when they awakened, because this was not considered in the study design.

Since there were no adverse reactions during the surgical procedure in the patients, we did not follow up the analysis of possible interactions with concomitant therapy, especially those drugs that were the most commonly administered together with midazolam, such as fentanyl, propofol, lidocaine, dexamethasone and acetaminophen.

We did not identify physiological limitations, given that the patients were ASA I and ASA II, who previously needed preoperative studies to ensure that they were fit to undergo surgery.

However, it is worth mentioning that the patients considered in the modeling had BMI values between the 10th and 90th percentiles according to the WHO and CDC tables, so they were considered eutrophic. In addition, in patients under 6 years of age, due to the immaturity of their metabolism, we observed that when administering the dose of 0.05 mg/kg, they remained deeply sedated until 120 min; therefore, we suggest that lower doses can be used in this group, so that in postoperative care, they do not take long to wake up from anesthesia.

## 5. Conclusions

The pharmacokinetic–pharmacodynamic behavior of midazolam at an IV dose of 0.05 mg/kg was determined in pediatric patients 2 to 17 years of age undergoing minor surgery to achieve deep sedation, using half the usual dose in this group of patients. Regarding age and sex differences, in <6-year-old patients, lower intravenous doses of 0.05 mg/kg midazolam may be required to achieve deep sedation, whereas in 6–17-year-old patients, they may require a second dose in surgeries lasting more than 120 min to remain deeply sedated. Therefore, adjusting the dosing regimen in these age groups is suggested, especially in male patients, who are likely to require a higher dose of sedation than recommended when undergoing minor surgeries.

In addition, further studies are needed to consider the effect of age and sex on the pharmacokinetics and pharmacodynamics of sedative agents, such as midazolam, because of their importance in pediatric anesthesiology therapeutics.

## Figures and Tables

**Figure 1 pharmaceutics-15-02565-f001:**
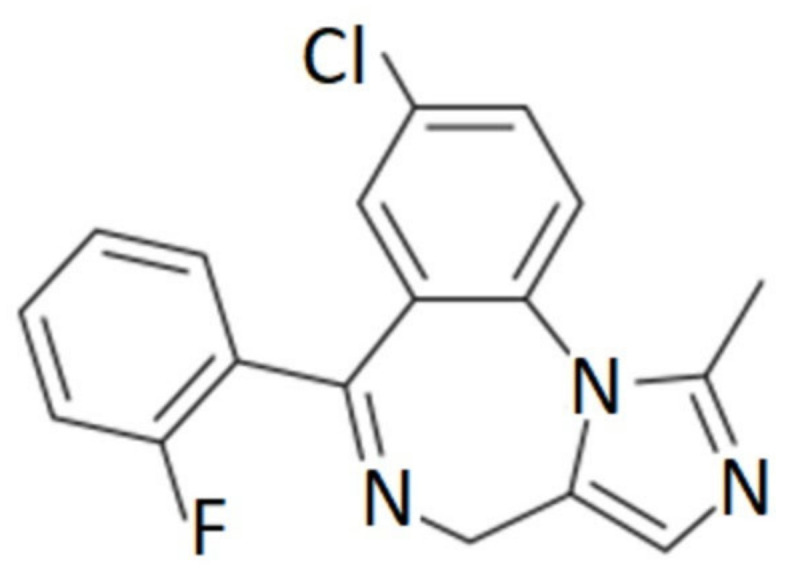
Chemical structure of midazolam.

**Figure 2 pharmaceutics-15-02565-f002:**
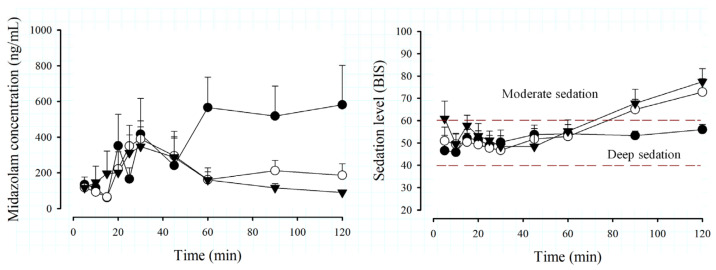
Time course of midazolam (MDZ) concentrations (left) and Biespectral Index (BIS) (right) by age groups, preschoolers, n = 26 (⬤); schoolchildren, n = 40 (◯); adolescents, n = 31 (▼), after administration of an intravenous dose of 0.05 mg/kg in patients scheduled for minor surgery. Data are expressed as the mean + SEM.

**Figure 3 pharmaceutics-15-02565-f003:**
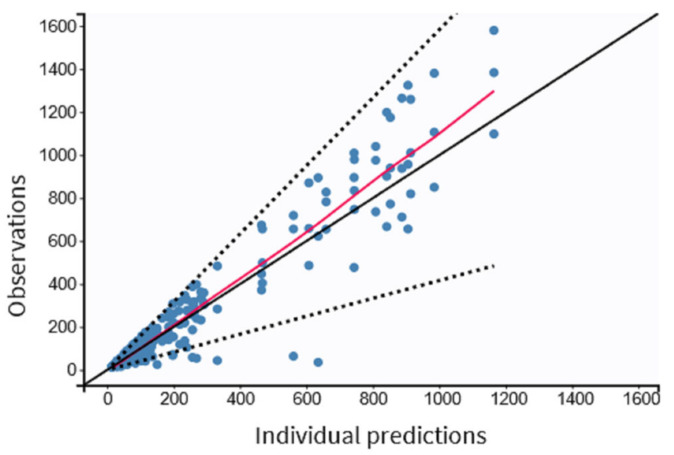
Observed versus predicted individual MDZ concentrations in pediatric patients undergoing minor surgeries (n = 97). Dashed lines correspond to the 90% prediction interval.

**Figure 4 pharmaceutics-15-02565-f004:**
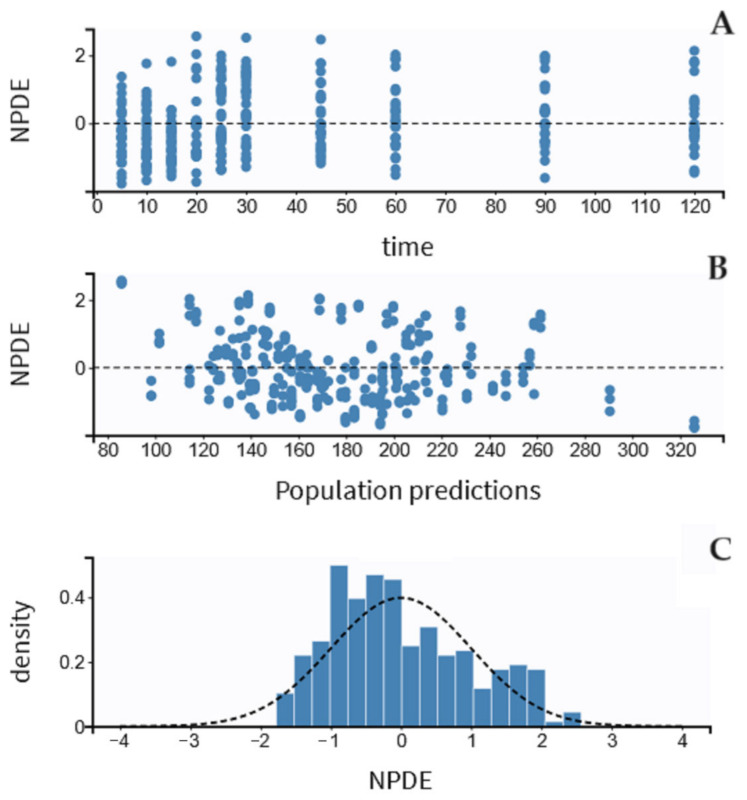
Results of the NPDE analysis for midazolam concentration: (**A**) the NPDE distributions over time, (**B**) against the observed concentrations, and (**C**) the NPDE histogram. NPDE indicates the normalized prediction distribution error, and the *Y*-axis indicates the probability density function.

**Figure 5 pharmaceutics-15-02565-f005:**
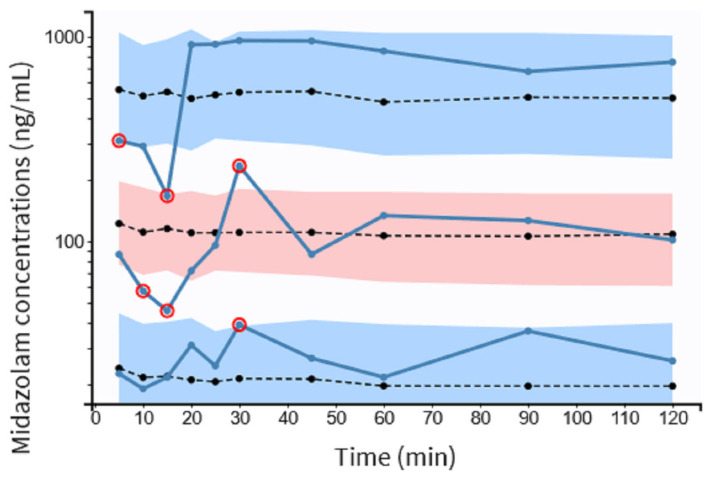
pc-VPC for midazolam concentrations versus time in minutes. The observed concentrations’ 5th, 50th, and 95th percentiles are represented by solid blue lines. Dotted lines represent the 5th, 50th and 95th percentiles of the observed concentrations. Colored areas represent the 95% confidence intervals for the 5th, 50th and 95th percentiles of simulated concentrations. The observed concentrations are shown by circles, red dots represent observations outside the confidence intervals. pc-VPC indicates the prediction-corrected visual predictive check.

**Figure 6 pharmaceutics-15-02565-f006:**
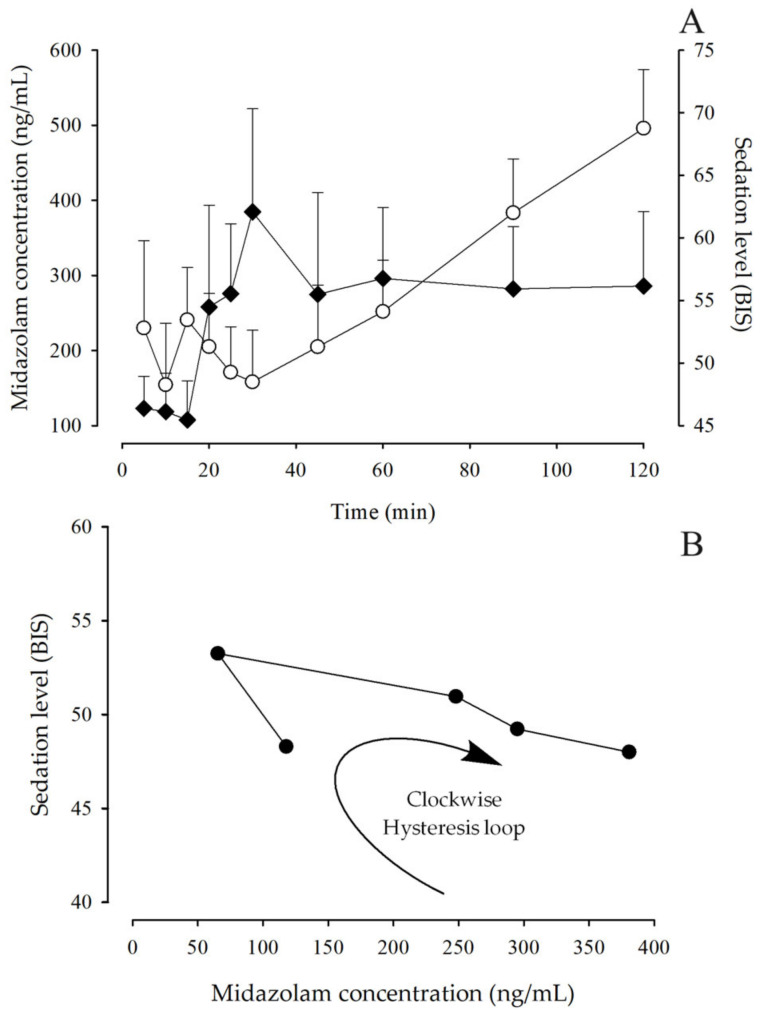
(**A**): Time course of MDZ blood concentrations (◆) and sedation level, expressed as BIS, (◯) after administration of an intravenous dose of 0.05 mg/kg in pediatric patients undergoing minor surgery (mean + SEM, n = 97). (**B**): Relationship between the observed sedation level and the measured blood concentration of MDZ; symbol (⬤) corresponds to the mean data taken from 10 to 60 min of the simulation.

**Figure 7 pharmaceutics-15-02565-f007:**
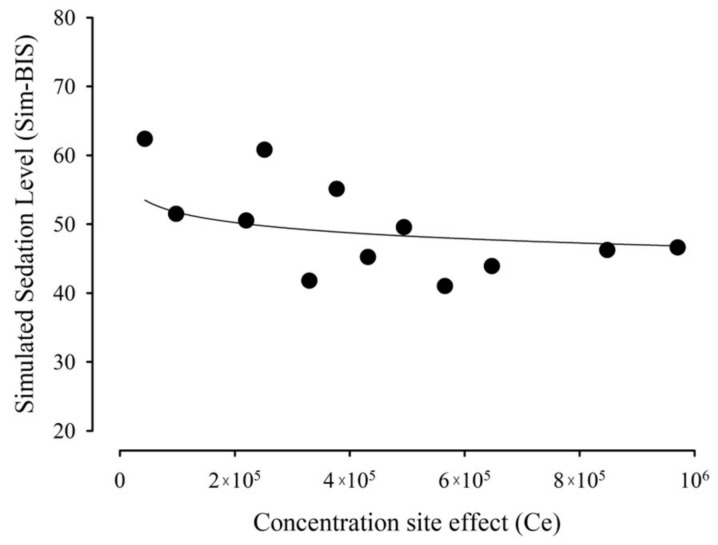
Relationship between the observed effect, measured as BIS, and calculated effect-compartment MDZ concentrations.

**Table 1 pharmaceutics-15-02565-t001:** Demographic and clinical characteristics of patients.

Characteristics	Total (n = 97)
Patients (n)	
Preschoolers	26
Schoolchildren	40
Adolescents	31
Sex (male/female)	
Preschoolers	19/7
Schoolchildren	22/18
Adolescents	18/13
Age (years) *	
Preschoolers	4 (3–5)
Schoolchildren	10 (8–12)
Adolescents	15 (14–17)
BMI (kg/m^2^) *	
Preschoolers	15.45 (13.9–17.3)
Schoolchildren	18.9 (16.1–21.8)
Adolescents	22.2 (19.75–24.1)
ASA n (%)	
I	67 (69.1%)
II	30 (30.9%)
Diagnostics n (%)	
Appendicitis	10 (10.3%)
Cryptorchidism	10 (10.3%)
Microtia	7 (7.2%)
Septal deviation	4 (4.1%)
Varicocele	3 (3.1%)
Breast fibroadenoma	3 (3.1%)
Cleft lip	2 (2.1%)
Others	58 (59.8%)

* Values expressed in median (interquartile range) Q_25_–Q_75_; Body Mass Index (BMI) expressed in values between the 10th and 90th percentiles = Eutrophic (WHO, CDC) [35,36].

**Table 2 pharmaceutics-15-02565-t002:** Population Pharmacokinetic Parameters.

Parameters	Value	RSE (%)	
Cl (mL/min)	0.48	148	
V1 (mL)	23.59	138	
Q (mL/min)	5091.62	147	
V2 (mL)	6792.42	16.3	
Standard Deviation of the Random Effects			
	Value	CV (%)	RSE (%)
ω^2^Cl (CV%)	2.66	3463.22	22.3
ω^2^V1 (CV%)	2.86	5971.61	32.3
ω^2^Q (CV%)	1.86	552.3	39.3
ω^2^V2 (CV%)	1.26	196.59	8.38

Cl, clearance; CV%, percentage coefficient of variation; Q, intercompartmental clearance; RSE, relative standard error; V1, central volume of distribution; V2, peripheral volume of distribution.

**Table 3 pharmaceutics-15-02565-t003:** Mean pharmacodynamic parameters for the *Imax* model.

	Value	RSE (%)	
*E*0	57.63	2.17	
*Imax*	0.088	0.036	
*IC*50	13.57	30.01	
Standard Deviation of the Random Effects			
	Value	CV (%)	RSE (%)
ω^2^*E*0 (CV%)	0.19	19.58	12.4
ω^2^*Imax* (CV%)	0.78	90.84	26.1
ω^2^*IC*50 (CV%)	2.99	8796.01	32.7

CV%, percentage coefficient of variation; *E*0, baseline effect; *IC*50, half-maximal inhibitory concentration; *Imax*, maximal fraction of inhibition; RSE, relative standard error.

## Data Availability

The data presented in this study are available upon request from the corresponding author or first author. The data are not publicly available due to ethical restrictions, as the information could compromise the confidentiality and privacy of research participants.

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
