# Peer review of "Pharmacokinetic–Pharmacodynamic Modeling of Midazolam in Pediatric Surgery"

_pharmaceutics, 2023, doi:10.3390/pharmaceutics15112565_

Round 1

Reviewer 1 Report

The study aimed to establish a PK/PD model for Midazolam in pediatric patients undergoing minor surgery to optimize dosing schedules and ensure adequate sedation without adverse effects.

The author should introduce in the manuscript the chemical structure of midazolam

A more detailed presentation of midazolam pharmacokineric, pharmacokinetic and pharmacotoxicologic profile should be presented in Introduction section

Questions:

1. Can you elaborate on the criteria used for patient selection, particularly concerning age and ASA status?

2. How were the three blood samples chosen randomly between 5-120 minutes after MDZ administration? What was the rationale behind this time frame?

3. From a practical standpoint, how do you envision these findings being applied in real-world clinical settings, especially concerning the administration of MDZ to pediatric patients?

4. What challenges did you encounter in fitting concentrations and BIS values to the sigmoid Emax PK/PD population and sigmoid Emax PK/PD indirect binding models?

5. What specific aspects of dosing schedules and adverse effects do you think need further exploration in future PK/PD studies?

6. What steps were taken to minimize bias and ensure the reliability of the observational data collected during the study?

  •  

The quality of English language is acceptable

Reviewer 2 Report

The reviewed article the results of an observational, longitudinal, prospective study of pharmacokinetic-pharmacodynamic effects of midazolam in pediatric surgery. The study is well conducted and presented, methods and results clearly exposed. From my point of view, since literature data is quite scarce this study adds important value to the subject. Also, one of the results, stating that younger than 6 old years population may require lower dose is of real value.

I have just one question, how was the study’s sample size and number of participants from each age group chosen? Also, a larger discussion about side effects would add some value

Reviewer 3 Report

The paper deals with an interesting experimental PK-PD study study about Midazolam in Pediatric Anesthesiology. The title is probably too generic, whereas the study is actually about the halving of the usual dose in some children pediatric populations.

We therefore suggest to be more to the point both in the title and introduction.

Despite laboratory methods having been described elsewhere, the manuscript would benefit from some experimental details. In particular, since both dried and liquid blood samples have been apparently used, a comparison of the data grouped by the type of samples used would be appreciated. This is mainly due to possible hematocrit related bias well known for dried blood samples. The hematocrit in children is usually very high and variable and potential biases are therefore expected.

In modelling and discussion, ethnic features should be emphasised to provide generality to important conclusion reached in the study.

Possible limitations due to concomittant therapies and physiological conditions limiting the application of the half dose should also be presented and discussed.

Albeit clear, the manuscript would definitely benefit from mother language editing due to over simplistic syntax and morphology

Reviewer 4 Report

It is a very interesting and meaningful study. The authors for the first time used the nonlinear mixed-effects, bicompartmental first-order elimination model to determine the pharmacokinetic-pharmacodynamic behavior of MDZ in a cohort of pediatric patients undergoing. This study gives clinicians and anesthetists an overall PK/PD relationship of MDZ in pediatric patients and its dose adjustment basis. The study was well designed and presented and should be accepted after some minor revisions.

1.      Could you the authors give more background introduction to PK/PD relationship of MDZ in adult patients? It would be better if the authors could give more comparisons of the difference in PK/PD between pediatric patients and adult patients.

2.      Duo to patients could be easily divided into two groups by their gender, could you the authors analyze the PK/PD relationship of MDZ in sex differences in pediatric patients?

3.      In Figure 1, we found that although the MDZ concentration was in the same concentration range in the distribution phase and elimination phase, there was an obvious difference in biespectral Index (BIS). Could you give some probable explanation for this phenomenon?

Minor editing of English language required

Round 2

Reviewer 1 Report

The article can be published in the current form

English is acceptable

Reviewer 3 Report

The revised version of the manuscript is ready for publication. In fact authors have answered all previous faults we signalled previously

English fine. Just a few minor adjustments to be addressed during proofreading